# A Predominant Cause of Recurrence of Ventilator-Associated Pneumonia in Patients with COVID-19 Are Relapses

**DOI:** 10.3390/jcm12185821

**Published:** 2023-09-07

**Authors:** Mirella van Duijnhoven, Manon Fleuren-Janssen, Frits van Osch, Jos L. M. L. LeNoble

**Affiliations:** 1Department of Intensive Care, VieCuri Medical Centre, Tegelseweg 210, 5912 BL Venlo, The Netherlands; m.w.h.janssen@viecuri.nl (M.F.-J.); jlenoble@viecuri.nl (J.L.M.L.L.); 2Department of Clinical Epidemiology, VieCuri Medical Centre, Tegelseweg 210, 5912 BL Venlo, The Netherlands; fvosch@viecuri.nl; 3Department of Epidemiology, NUTRIM, Maastricht University, 6200 MD Maastricht, The Netherlands; 4Department of Pharmacology and Toxicology, Maastricht University, 6200 MD Maastricht, The Netherlands

**Keywords:** COVID-19, mechanical ventilation, ventilator-associated pneumonia, recurrence, incidence, causative microorganism

## Abstract

The diagnosis of ventilator-associated pneumonia (VAP) recurrence in patients with coronavirus disease 2019 (COVID-19) pneumonia is challenging, and the incidence of recurrence is high. This study aimed to investigate the incidence and recurrence of VAP. Furthermore, we investigated the causative microorganisms of VAP and recurrent VAPs in patients with COVID-19. This retrospective, single-centre case series study was conducted during the COVID-19 pandemic from October 2020 to June 2021 at VieCuri MC Venlo. VAP and recurrent VAP were defined based on three criteria (clinical, radiological, and microbiological). During the study period, 128 mechanically ventilated patients with COVID-19 were included. The incidence ranged from 9.2 to 14 VAP/1000 ventilator days, which was higher than that in the non-COVID-19 controls. The most commonly cultured microorganisms in VAP were *Pseudomonas* (9/28; 32%), *Klebsiella* (8/28; 29%), *Escherichia coli* (5/28; 18%), and *Staphylococcus aureus* (5/28; 18%). VAP recurred often (5/19, 26%). The overall VAP rate confirmed previous findings of an increased incidence of VAP in critically ill patients with severe COVID-19 requiring mechanical ventilation. VAP recurrences occur often and are mainly relapses. A duration of antibiotic therapy longer than 7 days and therapeutic drug monitoring should be considered for VAP caused by Gram-negative microorganisms.

## 1. Introduction

The coronavirus disease 2019 (COVID-19) pandemic caused 769.369.823 confirmed cases worldwide, including 6.954.336 deaths by 10 August 2023 (WHO Corona Dashboard, www.who.it, accessed on 10 August 2023). Severe acute respiratory syndrome coronavirus-2 (SARS-CoV-2) is the causative agent of COVID-19. A large percentage (17–32%) of hospitalised patients for SARS-CoV-2 infection require admission to the intensive care unit (ICU), and 80% of these patients require invasive mechanical ventilation for a long period of time. These patients are at an increased risk of developing bacterial superinfections, including ventilator-associated pneumonia (VAP), which may contribute to an unfavourable prognosis [1,2]. VAP is associated with increased mortality, increased duration of mechanical ventilation, and ICU stay. VAP incidence in non-COVID-19 ICU patients receiving mechanical ventilation for more than two days ranges from 5% to 40%, while VAP incidence in patients with COVID-19 is higher, ranging from 50% to 80% [3,4]. In a multicentre cohort study, the incidence rate of VAP in patients with COVID-19 was 18 VAPs per 1000 ventilator days, which is in the higher range compared to the incidence rate of 1–19 VAPs per 1000 ventilator days in non-COVID-19 patients [5]. Potential explanations for the high incidence of VAP in patients with COVID-19 include the long duration of invasive mechanical ventilation, high incidence of acute respiratory distress syndrome (ARDS), and immunosuppressive treatment. Specific risk factors for VAP, such as bacterial–viral interactions in the lung microbiota, may also play a role in the pathogenesis of VAP [1,2,5,6,7].

VAP can be challenging to diagnose, especially in patients with COVID-19, as radiographic infiltrates, systemic inflammation, and impaired oxygenation are already present and may mimic the clinical picture that typifies VAP [1,8]. Recurrence rates of VAP are high in mechanically ventilated patients with COVID-19 and affect 40% of patients after the first episode of VAP [3]. This may have resulted in additional days of antibiotic treatment or mechanical ventilation. The risk factors for VAP recurrence include fever and mechanical ventilation on day 8 and ARDS, which increase the duration of invasive mechanical ventilation and consequently increase the risk of developing recurrent pulmonary infection [9]. Frequently, the growth of Gram-negative bacteria has been shown in multiple studies, with a high rate of *Pseudomonas aeruginosa* infection, while the outcome of patients with more than two episodes of VAP due to Pseudomonas infection is extremely poor [1,5,7].

The primary objective of this study was to assess the incidence and recurrence rates of VAP in mechanically ventilated patients with COVID-19 pneumonia. The secondary objective was to assess the results of cultures obtained from respiratory specimens during the first episode and VAP recurrence.

## 2. Materials and Methods

### 2.1. Study Design

This single-centre, observational, retrospective study was conducted between October 2020 and June 2021 at VieCuri MC Venlo, Netherlands. VieCuri MC is a large, non-university teaching hospital. Data were collected from patients’ medical records in a digital patient database and management system (HIX^TM^, Chipsoft, The Netherlands). This study was approved by the Medical Ethics Committee which waived the need for informed consent because of its retrospective nature (2022_074).

During the COVID-19 pandemic, the availability of ICU beds increased to 32. The relative number of confirmed COVID-19 cases in this region was the highest in the Netherlands. This study included all consecutive patients with COVID-19 aged ≥18 years, confirmed with a positive RT-PCR test on SARS-CoV-2 in respiratory specimens, who were admitted to the ICU and invasively ventilated. Lung protective ventilation (tidal volume 6 mL/kg, plateau pressure < 30 cm H_2_O, driving pressure < 15 cm H_2_O, and prone positioning when necessary) was applied to all invasively ventilated patients. Pressure control ventilation was the initial setting, which was, according to our weaning protocol, switched to pressure support if the patient was improving. When patients fulfilled the criteria of our weaning protocol (consisting of PEEP ≤ 8 cm H_2_O, PS ≤ 8 cm H_2_O, FiO_2_ ≤ 40%, hemodynamically stable, GCS > 8, and able to cough), a spontaneous breathing trial of 30 min was performed and when successful, the patient was extubated. HFNC or nasal oxygen was instituted after extubation. If a patient had multiple admissions, only the first was included in the analysis. No additional exclusion criteria were applied. We used data from the local registries of the VAP. These data were collected at our institution for epidemiological and other reasons. Our ICU also conducts complication registration, including registration for the diagnosis of VAP. During daily visits and multidisciplinary consultations, each invasively ventilated patient was examined to fulfil the criteria for diagnosing VAP. Measures for preventing VAP used in our ICU include daily interruption/decreasing of sedation, daily assessment of readiness to extubate, change of ventilator circuit only if visibly soiled, elevation of the head of the bed to 30–45 degrees (prone positioning was also applied), regular oral care with chlorhexidine, stress ulcer prophylaxis, and monitoring of residual gastric volumes. Our ICU team did not perform selective decontamination of the digestive tract (SDD). During the COVID-19 pandemic, personal protective equipment (full-length fluid impermeable gowns, FFP2 masks, gloves, and glasses) was used, and one-to-one nursing-to-patient ratios were maintained.

Bacterial cultures were obtained with endotracheal aspiration. A catheter was inserted via the endotracheal tube into the trachea, and when inserted to the appropriate depth, intermitted suction was applied while withdrawing the catheter. The specimen was collected in a sterile container. In our hospital, semiquantitative cultures are performed. The clinical microbiological laboratory in our hospital is under the surveillance of a quality management system, accredited by ISO 15189, using standard operating procedures and according to the national “Workgroup Infection Prevention” guidelines. Standard culture plates, which are periodically checked, are used and determination of microorganisms is performed via MaldiTof.

According to national and international recommendations, empiric antibiotic therapy was started in case of VAP suspicion. Antibiotic stewardship was performed with daily visits by a pharmacist and a clinical microbiologist. Antibiotic treatment and dosage were adjusted according to patient-related factors and culture results. In our hospital, VAP is treated for seven days.

### 2.2. Data Definition

The ventilator days of both VAP and non-VAP patients with COVID-19 were collected to calculate the incidence rate of VAP. VAP was defined as new or worsening chest X-ray infiltrates occurring more than 48 h after the initiation of invasive mechanical ventilation and a positive respiratory secretion culture with a pathogenic microorganism plus both of the following: (1) new onset of fever (body temperature > 38 °C) and/or leucocytosis (<3.5 × 10^9^/L or >11 × 10^9^/L), and (2) new onset or change in respiratory secretions and/or worsening of oxygenation. The primary endpoint of the study was the incidence of VAP expressed per 1000 ventilator days.

VAP recurrence was defined as new-onset VAP following the regression of clinical signs and inflammatory biomarkers after complete antibiotic treatment. VAP recurrence was diagnosed on the basis of the same clinical, radiological, and microbiological criteria as the first episode of VAP. A distinction between recurrence and relapse was made based on microbiological signals. VAP recurrence was defined as the bacterial growth of microorganisms different from the first episode of VAP. Relapse was defined as the recurrence of at least one of the initial causative microorganisms [7].

### 2.3. Data Collection

Data collected included demographic characteristics, duration of ICU admission, date of intubation, duration of mechanical ventilation received during ICU admission, and endotracheal aspiration specimens. Medication data included dosages and durations of dexamethasone, tocilizumab, and antibiotics. These results were compared with those of a non-COVID population that was mechanically ventilated in our ICU from 1 January 2019 to 31 December 2019.

### 2.4. Statistical Analysis

All continuous data are reported as medians with interquartile ranges (IQR) and categorical data as numbers and percentages. Baseline patient characteristics were compared between the VAP and non-VAP groups using the Chi-square or Fisher’s exact test for categorical variables and the Mann–Whitney U test for non-parametric continuous variables. Data analyses were performed using IBM SPSS Statistics version 26 (IBM Corp., Armonk, NY, USA). A two-tailed *p*-value of ≤0.05 was considered significant in all analyses.

## 3. Results

### 3.1. Patients

During the study period, 190 patients with COVID-19 were admitted to our ICU, of whom 62 were excluded due to the absence of invasive mechanical ventilation. In total, 128 eligible patients were included in the present study, with a median age of 65 years and 30% females (38/128) (Table 1).

Dexamethasone (1 dd 6 mg for 10 days) was administered to all patients, and ceftriaxone or other broad-spectrum antibiotics were administered on the first day of ICU admission to 82 patients (82/128; 84%). Tocilizumab (8 mg/kg) was administered to 25% of patients (32/128; 25%). Treatment protocols for COVID-19 were evolving during the pandemic, and tocilizumab was added to the protocol during the study period, explaining the low number of patients receiving tocilizumab.

### 3.2. Invasive Mechanical Ventilation

All patients were invasively ventilated for more than 48 h with a total of 2.072 ventilator days (mean duration of mechanical ventilation: 16 d; range 1–61 d of mechanical ventilation). Eighteen definite VAPs and ten probable VAPs occurred during the study period, with an incidence of 9.2–14/1000 ventilator days. This incidence is considerably increased compared with the historical data of non-COVID-19 ICU patients, where a total of nine VAPs were diagnosed in 1.680 ventilation days (5.4/1000 ventilator days). Duration of mechanical ventilation, length of ICU stay, and mortality were negatively affected by VAP.

### 3.3. VAP Recurrence

VAP recurrence occurred in five patients (5/19; 26% (95% CI = 9–51%), with an incidence of 40/1000 ventilator days and a range–2–5 recurrences per patient (Figure 1 and Table 2).

There were two recurrences and seven relapses. Of these seven relapses, two were seen as part of the first episode of VAP because the duration of treatment was less than seven days when cultures were repeated (Table 2). No data were available on the recurrence of VAP during the pre-COVID-19 period.

The first episode of VAP was diagnosed by the attending physician 13 days after initiating invasive mechanical ventilation (Table 2). Recurrence/relapse of VAP was diagnosed four times on day 7 after the first episode of VAP and once after 15 days (Table 2).

### 3.4. Pathogens in VAP

The results of the cultures with different pathogens in VAP are shown in Table 2. The most commonly cultured microorganisms in VAP were *Pseudomonas* (9/28; 32% (95% CI = 16–52%)), *Klebsiella* (8/28; 29% (95% CI = 13–49%)), *Escherichia coli* (5/28; 18% (95% CI = 6–37%)), and *Staphylococcus aureus* (5/28; 18% (95% CI = 6–37%)).

In all VAP recurrences or relapses, the same Gram-negative causative microorganisms (*Pseudomonas*, *Klebsiella* and *E. coli*) were found (Table 2). VAP treatment with antibiotics was adjusted according to the resistance patterns of the microorganisms. The duration of antimicrobial treatment for VAP or recurrence/relapse of VAP was maximal 7 days.

## 4. Discussion

This case series was designed to investigate the incidence and causative microorganisms of recurrent VAP in patients with COVID-19. We found an increased incidence of VAP in critically ill patients with COVID-19 (9.2-14/1000 ventilator days) compared to that in non-COVID-19 patients from a similar cohort (5.4/1000 ventilator days). We also demonstrated that VAP recurrence was high in patients with COVID-19 (40/1000 ventilator days), occurring in 26% (in 5 of 19 patients with VAP; (95% CI = 9–51%)) after the first episode of VAP, and was mostly due to a relapse (caused by the same microorganism as the first episode of VAP). Although this was a case series study, our results suggest that VAP has a negative effect on the duration of mechanical ventilation, length of ICU stay, and mortality, as previously described [1,2,5,6,7].

These results confirm those of previous studies showing a higher incidence rate of VAP in patients with COVID-19 [1,2,5,6,7]. This could be explained by the pathogenesis of the viral infection. SARS-CoV-2 binds to the acetylcholine-2 receptor which is present in pulmonary tissue and leads to the internalisation of the virus into alveolar cells. Virus replication in alveolar cells mediates damage and induces an inflammatory response by releasing cytokines and interferons. This damage leads to ARDS, a risk factor for VAP development. The additional immunosuppressive effect of the deep lymphopenia caused by the virus, presumably caused by the release of interferons, and treatment with immunosuppressive agents (dexamethasone and tocilizumab (an IL-6 receptor monoclonal antibody)) [10,11,12] lead to the predisposition of bacterial superinfection. However, VAP incidence did not increase after instituting dexamethasone as a treatment for COVID-19 infection and was not associated with an increased duration of mechanical ventilation [11]. In another study, in patients with COVID-19 treated with dexamethasone, VAP also occurred earlier (on day 7 instead of day 13) [13]. In this study, all patients were treated with dexamethasone, and VAP occurred on day 13. To our knowledge, whether the addition of tocilizumab to the treatment regimen resulted in an increase in VAP incidence in patients with COVID-19 has not been investigated. Also, the effect of SDD on the incidence of VAP in patients with COVID-19 is still unknown. The incidence of VAP in hospitals that have implemented SDD has not yet been assessed.

In our study, VAP incidence was high compared to that in non-COVID-19 patients. Recurrence or relapse of VAP occurred in five patients (5/19 patients with VAP; 26%; (95% CI = 9–51%). However, in two patients, a relapse of VAP was diagnosed within 7 days after the diagnosis of the previous VAP. Whether this was a relapse or a new episode of VAP remains unclear. Endotracheal aspiration is performed when patients do not recover clinically or persistently produce purulent sputum. Antibiotic treatment was adjusted according to the resistance pattern of the microorganisms found in the cultures as soon as possible to decrease the chance of microorganisms becoming resistant.

The results of our study are consistent with those of previous studies showing that *Pseudomonas*, *Klebsiella*, *E. coli*, and *S. aureus* are the most common causative microorganisms of VAP in patients with COVID-19 [1,2].

Relapse of VAP is mostly caused by Gram-negative microorganisms. The duration of antimicrobial therapy in these patients was 7 days. Generally, the duration of VAP treatment is 7 days, which does not result in an increase in the number of VAP relapses compared to a longer duration of treatment. However, this is not applicable to non-fermenting Gram-negative bacilli, for which a short course can lead to a higher VAP relapse rate [14]. In COVID-19 pneumonia, VAP recurrence rates are high (40% of patients with VAP had at least one recurrence; in the normal IC population, the recurrence rate ranged from 8% to 25% [3]), although the treatment duration was 7 days. This is probably due to the pulmonary inflammation caused by COVID-19, which makes the lung parenchyma more prone to bacterial superinfection. Furthermore, non-fermenting Gram-negative bacilli possess multiple virulence mechanisms that enable colonisation and subsequent tissue invasion [15]. A longer duration of treatment, especially for VAP caused by Gram-negative microorganisms, should be considered, which has been confirmed by a Cochrane review addressing multiple studies [14]. The COVID-19 pandemic led to increases in antimicrobial resistance in the ICU setting, indicating the importance of antibiotic stewardship programmes and infection control measures [16]. This is an important factor in the treatment of VAP. Furthermore, therapeutic drug monitoring (TDM) should be considered. Homeostatic changes due to disease processes and required interventions on top of chronic co-morbidity are present in critically ill patients. This leads to alterations in pharmacokinetic and pharmacodynamics processes. The distribution volume is altered due to endothelial dysfunction leading to capillary leakage, and renal clearance is often altered in critically ill patients. In the ICU, bacterial pathogens may demonstrate higher minimal-inhibitory concentration (MIC) when compared to general wards. These variabilities can lead to sub-optimal antibiotic exposure, and TDM is a safe and effective way to ensure the achievement of therapeutic antimicrobial exposure [17]. For example, higher dosages and longer exposures of beta-lactam antibiotics (i.e., piperacillin/tazobactam) or fluoroquinolones (i.e., ciprofloxacin), which are used in our hospital for the treatment of Gram-negative microorganisms, may be beneficial for critically ill patients [17]. Currently, the Dutch national guidelines for antibiotic therapy “Stichting Werkgroep Antibiotica Beleid” (SWAB) recommend piperacillin/tazobactam in a fixed dosage of 4500 mg four times a day for five days, extended to seven days only in case of sepsis, while international guidelines (European Society of Intensive Care Medicine (ESICM) and Infectious Diseases Society of America (IDSA)) recommend a therapy duration of a minimum of seven days. The IDSA also recommends pharmacokinetic and pharmacodynamic optimised dosing by using antibiotic blood concentration, extended and continuous infusion, and weight-based dosing when applicable [18,19]. 

Further studies are needed to investigate the continuum of VAP and relapse and recurrence of VAP, especially in COVID-19 patients. Improvements in the criteria for diagnosing VAP and relapse/recurrence of VAP are needed. Also, additional preventive measures, such as SDD and fine-tuning of treatment options such as TDM, should be investigated.

This study had some limitations. First, due to the relatively small sample size, this study could only show large effect sizes (>0.7 or proportion differences larger than 33%) while comparing patients with VAP to patients without VAP. Furthermore, there were no ICU admissions of non-COVID-19 patients during the study period, which made a case-control study impossible. Second, this was a single-centre, observational, retrospective study, which implies a risk of bias. The analysis was performed by first selecting the endotracheal aspirate results instead of clinical suspicion or signs of VAP. Both can lead to selection bias. Third, diagnosing VAP in COVID-19 pneumonia is challenging because many features of VAP, such as pulmonary X-ray abnormalities, fever, elevated inflammatory markers, and hypoxaemia, are similar to those of COVID-19 pneumonia. No quantitative cultures were used, and it was difficult to distinguish between colonisation and infection. Furthermore, not all data were noted in the electronic patient files, which complicated the diagnosis of VAP and led to information bias. In addition, owing to the interhospital transport of patients, follow-up was lost. This may have biased the results.

## 5. Conclusions

The incidence rate and recurrence of VAP are higher in patients with COVID-19 compared to those in non-COVID-19 patients and are mainly relapses. However, the causative microorganisms were mostly comparable to those in non-COVID-19 patients with VAP. A longer duration and a higher dosage of antimicrobial treatment with therapeutic drug monitoring should be considered, especially when VAP relapse(s) occurs because of Gram-negative microorganisms.

## Figures and Tables

**Figure 1 jcm-12-05821-f001:**
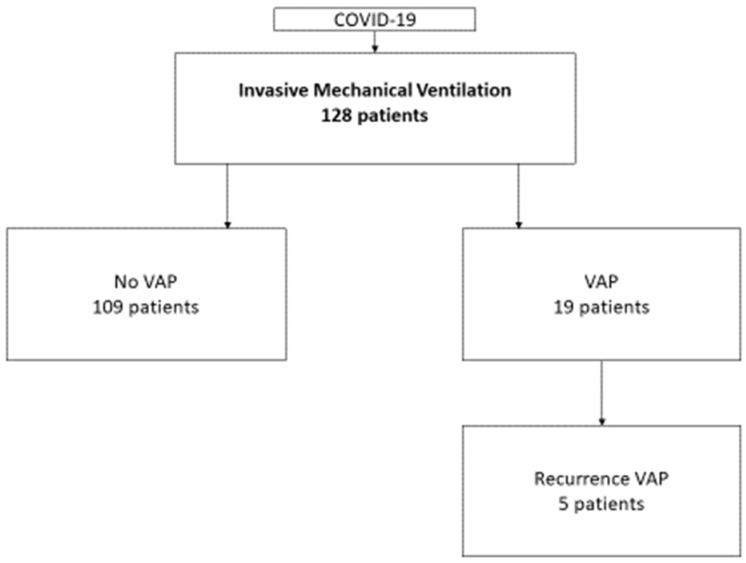
Patient selection flowchart: Number of invasively ventilated patients with COVID-19 with no ventilator-associated pneumonia (VAP) or with VAP and recurrence of VAP.

**Table 1 jcm-12-05821-t001:** Characteristics and demographics of included patients with and without ventilator-assisted pneumonia (VAP).

Patient Characteristics & Demographics	Total (n = 127)	No VAP (n = 108)	VAP (n = 19)	*p*-Value ^1^
Age ^2^	68 (45)	66 (45)	72 (36)	0.04
Female, n (%)	37 (29)	34 (32)	3 (16)	0.27
ICU length of stay ^2^	14 (80)	11 (62)	41 (80)	<0.001
Duration of mechanical ventilation ^2^	10 (79)	9 (52)	32 (74)	<0.001
In-hospital mortality, n (%)	39 (31)	29 (27)	10 (53)	0.03
Co-morbidities				
Overweight (BMI > 25), n (%)	111 (87)	96 (89)	15 (79)	0.26
Diabetes Mellitus, n (%)	37 (29)	32 (30)	5 (26)	0.79
COPD, n (%)	27 (21)	25 (23)	2 (11)	0.36
Chronic kidney disease, n (%)	10 (8)	10 (9)	0	0.36
Treatment				
Antibiotics < 24 h after admission, n (%)	82 (64)	70 (65)	12 (63)	0.88
Dexamethasone (6 mg; 10 days), n (%)	127 (99)	107 (99)	19 (100)	1.00
Tocilizumab (8 mg/kg), n (%)	29 (23)	27 (25)	2 (11)	0.24

^1^ Mann–Whitney U test for continuous variables and Fisher’s exact test for categorical variables. ^2^ Presented as median (interquartile range). ICU, intensive care unit; BMI, body mass index; COPD, chronic obstructive pulmonary disease.

**Table 2 jcm-12-05821-t002:** Ventilator-associated pneumonia (VAP) episodes and recurrence/relapse in patients with COVID-19 with culture results.

Patient	VAP Episode	Definite/Probable VAP	Recurrence/Relapse	Endotracheal Aspirate	Ventilator Days before VAP
1	1st	Definite	-	*E. coli*; *Klebsiella*	9
2	1st	Definite	-	*Serratia*	5
3	1st	Definite	-	*Klebsiella*; *E. faecium*	26
4	1st	Definite	-	*Pseudomonas*	16
5	1st	Definite	-	*Pseudomonas*	44
6	1st	Definite	-	*Pseudomonas*	8
	2nd *	Definite	Relapse *	*Pseudomonas*	12
	3rd	Probable	Relapse	*Pseudomonas*	19
7	1st	Definite	-	*S. aureus*	6
	2nd	Definite	Recurrence	*Klebsiella*	13
	3rd	Definite	Relapse	*Klebsiella*	20
	4th	Definite	Relapse	*Klebsiella*	28
	5th *	Definite	Relapse *	*Klebsiella*	30
8	1st	Definite	-	*S. aureus*; *Bacillus cereus*; *C. albicans*	9
9	1st	Definite	-	*S. aureus*	6
10	1st	Definite	-	*E. coli*	22
11	1st	Definite	-	*Serratia*	9
12	1st	Probable	-	*Pseudomonas*; *Klebsiella*; *S. aureus*	10
	2nd	Definite	Relapse	*Pseudomonas* (2 species); *Klebsiella*	17
13	1st	Definite	-	*E. faecalis*; *S. aureus*	6
14	1st	Probable	-	*S. pneumonia*	9
15	1st	Probable	-	*E. faecalis*; *C. albicans*	9
16	1st	Probable	-	*E. coli*	9
	2nd	Probable	Recurrence	*E. cloaca*	24
17	1st	Probable	-	*E. cloaca*	8
18	1st	Probable	-	*Pseudomonas*	19
19	1st	Probable	-	*E. coli*; *C. albicans*	16
	2nd	Probable	Relapse	*E. coli*; *C. albicans*	23
Mean ventilator days before 1st VAP					13

VAP occurred in patients with COVID-19, with the first episode occurring at an average of 13 days after the initiation of invasive ventilation. Causative microorganisms found in the endotracheal aspirate are shown. There were VAP recurrences in five patients, of which two relapses were dubious because they were diagnosed during antibiotic treatment for VAP. In patient 12, two different species of Pseudomonas species were found. * Analysed as relapse; however, cultures were obtained within 7 days of antibiotic treatment for the previous episode of VAP.

## Data Availability

All the data from this study are available from the corresponding author upon request.

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
