# Peer review of "A Predominant Cause of Recurrence of Ventilator-Associated Pneumonia in Patients with COVID-19 Are Relapses"

_jcm, 2023, doi:10.3390/jcm12185821_

Round 1

Reviewer 1 Report

This article presents a systematic study of the incidence and recurrence rates of ventilator-associated pneumonia (VAP) in COVID-19 patients, contributing significantly to the understanding of clinical manifestations and impacts of VAP in this context. The structure of the article, including the introduction, methods, and results, is well-organized, providing a clear presentation of the research objectives, methods, and findings. Below is an evaluation and suggestions regarding your article:

1. Introduction:

The introduction effectively highlights the association between COVID-19 and VAP, as well as the potential reasons for increased susceptibility to VAP in COVID-19 patients.

Consider incorporating additional background information in this section, such as the global prevalence of the COVID-19 pandemic and previous research findings on VAP incidence in COVID-19 patients, to provide readers with more comprehensive context.

2. Materials and Methods:

You adequately describe the research design, data collection, and analysis methods, enabling readers to understand the study process.

In the "Data Collection" subsection, you could provide further details on how respiratory specimens were collected, bacterial culture methods, and the steps involved in isolating and identifying pathogenic microorganisms.

3. Results:

You present valuable data on the incidence and recurrence rates of VAP in COVID-19 patients, as well as the bacterial culture results. These findings contribute to a better understanding of disease characteristics and trends.

In the "Discussion" section, you could delve deeper into the analysis of your results, exploring why COVID-19 patients are more susceptible to VAP and potential mechanisms behind the higher recurrence rates. Additionally, you could compare your research results with similar studies to further validate your findings.

4. Discussion:

In the "Discussion" section, you can provide more in-depth explanations of your results, discussing their clinical and epidemiological significance, as well as exploring the impact of VAP on the recovery and treatment of COVID-19 patients. You could also offer suggestions on how to improve prevention and treatment strategies to reduce VAP incidence and recurrence.

5. Conclusion:

In the conclusion section, you can summarize your research findings and emphasize their importance in the management of COVID-19 patients. You may also propose directions for future research to address existing knowledge gaps.

In summary, your article furnishes crucial insights into the incidence and recurrence rates of VAP in COVID-19 patients. However, the discussion section could be further enriched to offer deeper analysis and insights, along with practical implications and recommendations for clinical practice. Furthermore, ensure that your citations and references adhere to the journal's requirements, enhancing the overall acceptability of the article.

Minor editing of English language required

Author Response

18 August 2023

Dear reviewer of The Journal of Clinical Medicine,

Thank you for your comments and suggestions to our article “A predominant cause of recurrence of Ventilator-Associated Pneumonia in patients with COVID-19 are relapses”. We have adjusted our article and the subsequent changes are indicated in red. 

Additional background information about the COVID-19 pandemic and research findings on VAP in COVID-19 patients has been incorporated into the introduction. 

The Material & Methods section is extended with further details about respiratory specimens’ collection and culture methods under the subsection “Study Design”. We’ve discussed whether to place it under subsection “Data collection”. However, in our humble opinion it belongs to the “Study Design” section.  Our microbiological laboratory follows the quality management system (ISO 15189) rules and national guidelines formulated by the “Workgroup Infection Prevention”, which we also added to the Material & Methods section to further specify the workflow in our microbiological laboratory.

In the Discussion section we added more in-depth explanations of the pathogenesis of COVID-19, which could explain the higher rate of VAP and higher incidence of recurrence of VAP. Measures to prevent VAP, as mentioned in the Material & Methods section and according to (inter)national guidelines, where continued in our ICU during the pandemic and we deployed additional personal protective equipment. Unfortunately, this didn’t result in a similar or decreased VAP incidence in our patients with COVID-19 compared to patients without COVID-19. SDD could be a protective prevention strategy. However, in our hospital SDD is not instituted and to our knowledge, no studies comparing SDD to standard treatment in prevention of VAP in patients with COVID-19 are conducted yet. The treatment strategies to reduce VAP incidence and recurrence are further specified and described in more detail with future suggestions in the Discussion section.

Our paper has been reviewed by “Editage” (www.editage.com), a recommended editage-service for articles written in English. A copy of the confirmation certificate has been sent to the editorial office. 

We also adjusted our citations and references to the journal requirements using EndNote.

Hopefully, our adjustments comply with your comments and suggestions and are sufficient for consideration to publish our article in The Journal of Clinical Medicine. 

Sincerely,

Mirella van Duijnhoven

Department of Intensive Care, VieCuri Medical Centre, Tegelseweg 210, 

5912 BL Venlo, The Netherlands

Tel.: +31-77-3205392

[email protected]

Reviewer 2 Report

The aim is clear, methods are ok.

I have only some comments:

1) It should be important to better describe the ventilator settings and weaning protocol of the center, including the use of CPAP/NIV for weaning protocol and use of HFNC after extubation.

2) Please provide a post-hoc sample size computation, to assess the power of the study (and to understand if the sample is or not underpowered)

3) It should be interesting to discuss your findings in the light of other reports (for example PMID: 34373952 and 35630400 ).

Author Response

18 August 2023

Dear reviewer of The Journal of Clinical Medicine,

Thank you for your kind words as well as your comments and suggestions. We adjusted our article “A predominant cause of recurrence of Ventilator-Associated Pneumonia in patients with COVID-19 are relapses”. Changes are indicated in red.

In the Material&Methods section, we describe the ventilator settings and weaning protocol of our centre.

Also, a post-hoc sample size computation is done and added to the results. When comparing the 19 VAP patients to the 108 patients without VAP we were able to show effect sizes of >0.7 in continuous variables such as age between these groups. For categorical variables such as diabetes mellitus in the Fisher’s exact test, the difference in proportion that could have been shown would have to be at least 33% (no statistically significant differences were observed between the groups, since all were smaller than 20%). The recurrence of VAP was only described (5/19 patients). But we have added a 95% CI to further clarify the uncertainty that surrounds the estimate of 26% (9.1%-51.2%) to the manuscript file and other estimates for pathogens in VAP.

Thank you for your suggestions for discussing our results in light of the mentioned reports. We added one of these articles to our discussion.

Hopefully, our adjustments comply with your comments and suggestions and are sufficient for consideration to publish our article in The Journal of Clinical Medicine.

Sincerely,

Mirella van Duijnhoven

Department of Intensive Care, VieCuri Medical Centre, Tegelseweg 210,

5912 BL Venlo, The Netherlands

Tel.: +31-77-3205392

[email protected]